# Decrease in decision noise from adolescence into adulthood mediates an increase in more sophisticated choice behaviors and performance gain

**Vanessa Scholz**[1,2]*, **Maria Waltmann**[1,3], **Nadine Herzog**[3,4], **Annette Horstmann**[3,4,5], **Lorenz Deserno**[1,3,4,6]*

**1** Department of Child and Adolescent Psychiatry, Psychosomatics and Psychotherapy, Centre of Mental Health, University of Würzburg, Würzburg, Germany, **2** Donders Institute for Brain, Cognition and Behaviour, Radboud University, Nijmegen, the Netherlands, **3** Max Planck Institute for Human Cognitive and Brain Sciences, Leipzig, Germany, **4** IFB Adiposity Diseases, Leipzig University Medical Center, Leipzig, Germany, **5** Department of Psychology and Logopedics, Faculty of Medicine, University of Helsinki, Helsinki, Finland, **6** Department of Psychiatry and Psychotherapy, Technical University Dresden, Dresden, Germany

* scholz_v@ukw.de (VS); deserno_l@ukw.de (LD)

## Abstract

Learning and decision-making undergo substantial developmental changes, with adolescence being a particular vulnerable window of opportunity. In adolescents, developmental changes in specific choice behaviors have been observed (e.g., goal-directed behavior, motivational influences over choice). Elevated levels of decision noise, i.e., choosing suboptimal options, were reported consistently in adolescents. However, it remains unknown whether these observations, the development of specific and more sophisticated choice processes and higher decision noise, are independent or related. It is conceivable, but has not yet been investigated, that the development of specific choice processes might be impacted by age-dependent changes in decision noise. To answer this, we examined 93 participants (12 to 42 years) who completed 3 reinforcement learning (RL) tasks: a motivational Go/ NoGo task assessing motivational influences over choices, a reversal learning task capturing adaptive decision-making in response to environmental changes, and a sequential choice task measuring goal-directed behavior. This allowed testing of (1) cross-task generalization of computational parameters focusing on decision noise; and (2) assessment of mediation effects of noise on specific choice behaviors. Firstly, we found only noise levels to be strongly correlated across RL tasks. Second, and critically, noise levels mediated age-dependent increases in more sophisticated choice behaviors and performance gain. Our findings provide novel insights into the computational processes underlying developmental changes in decision-making: namely a vital role of seemingly unspecific changes in noise in the specific development of more complex choice components. Studying the neurocomputational mechanisms of how varying levels of noise impact distinct aspects of learning and decision processes may also be key to better understand the developmental onset of psychiatric diseases.

**Data Availability Statement:** Data and code for analyses and figures included in this manuscript are available publicly via https://osf.io/mcx36/.

**Funding:** This work was directly funded by a grant to L.D. and A.H. from the IFB Adiposity Diseases, Federal Ministry of Education and Research (BMBF: https://www.bmbf.de), Germany, GN: 01EO150. LD also receives funding from the German Research Foundation (DFG: https://www.dfg.de/) as part of the Collaborative Research Centre 265 (Project A02, Project Number: 402170461) and on ADHD (DE 2509/3-1, Project number 533682086) as well as by the BMBF on the computational foundations of internalizing versus externalizing symptoms (01GQ2302B), which partially supported this work. The funders had no role in study design, data collection and analysis, decision to publish or preparation of the manuscript.

**Competing interests:** The authors have declared that no competing interests exist.

**Abbreviations:** ADHD, attention-deficit hyperactivity disorder; HBI, hierarchical Bayesian inference; MB, model-based; PXP, protected exceedance probability; RL, reinforcement learning.

## Introduction

Learning and decision-making change considerably from adolescence into adulthood [1–3]. Adolescents have to learn how to navigate a constantly changing environment and to strike a balance between exploring new outlets and sticking with already known "good" choices [4]. They do so while their brain undergoes substantial maturational changes that particularly affect cognitive control, value-based learning, and choice [3,5,6]. As yet, the neurocomputational processes accompanying these developmental challenges remain insufficiently understood [4]. Here, reinforcement learning (RL) models provide a computational framework to test hypotheses about latent processes underlying the development of learning and decision-making [7], e.g., learning rates or sensitivity to different outcomes, which are not accessible by the analysis of overt behavior.

A focus in the RL literature has been on the development of specific learning and choice signatures [3,8–18], especially during adolescence, a period frequently considered to be a "window of opportunity" for exploring new choice options and outlets [19]. For example, it was demonstrated that goal-directed behavior (model-based (MB) control over choice), increases from adolescence to adulthood [8–15]. Reward-based cognitive flexibility, as measured by reversal learning paradigms, improves with age. Another example is the influence of motivation on choices [17,20]. Individuals can be strongly biased to respond with behavioral activation to obtain reward, while punishment tends to facilitate behavioral inhibition [21–24]. This so-called Pavlovian choice bias has been found to be reduced in adolescence relative to childhood and adulthood (i.e., weaker behavioral activation to rewards and lower behavioral inhibition to punishments), which was suggested to result from an elevated exploration in adolescents [17].

In the recent past, it was noted that such specific developmental changes in learning and choice tend to be inconsistent across studies [25] and generalize rather poorly [4]. Likewise, some key parameters of RL models, e.g., the learning rate, did not generalize across tasks, which may reflect specific and necessary adaptation to experimental environments [4,26]. Slight differences in experimental environments (e.g., frequency of changes in reward contingencies) impact results, such that performance may be improved or hampered in youths as compared to older adults (compare results from reversal learning in [16] to [26]). On the other hand, one very consistent developmental finding across different tasks is that adolescents show increased levels of decision noise (or choice stochasticity [16,27], for review [4,28]). Decision noise describes a decoupling of learned values from action selection and leads to increased variability in choice behavior. This leads to the selection of less optimal or lower-valued options and usually does not optimize reward outcomes. However, such behavior could, in principle, be regarded as explorative [29]. Yet, in most tasks discussed so far, this noisy or random exploration cannot be clearly distinguished from directed information seeking. The latter aims at explicit information gain, a hallmark for more sophisticated styles of exploration [30]. Thus, decision noise has often been regarded as less interpretable, potentially even reflecting incompliance to experimental designs.

One unexplored possibility is that the development of specific and more sophisticated choice behaviors may be related or even depend on individual levels of decision noise. To study this unaddressed question, the central aim of this study was to shed light on the role of developmentally elevated decision noise within and across 3 RL tasks. We collated and (re)analyzed data from a developmental sample that had completed the 3 RL tasks as part of a larger study protocol [8,16,23,31–33]. We set the stage by replicating specific developmental effects from a motivational Go/NoGo task capturing decision biases. Using a modified implementation of noise in an RL model of the motivational Go/NoGo task, we show that, in line with

previous work, decision noise may reflect a consistent (rather stable within-subject) signature characteristic for an individual [26]. Critically, we then set out to test our main novel proposal of mediating effects of such seemingly unspecific decision noise on developmental changes in specific choice signatures and corresponding performance readouts from all 3 RL tasks. We hypothesized that noise levels might mediate the relationship between age and specific developmental task effects. This expectation was based on previous findings of developmental changes in noise levels [26,34] as well as evidence suggesting that noise undermines core cognitive processes such as information updating and integration, both crucial for learning and decision-making [35].

## Results

We analyzed data from 93 participants between 12 and 42 years of age (age mean [SD] = 22.65 [7.88], female: 45; male: 48) who completed 3 reinforcement learning tasks as part of a larger protocol [36]: (1) a modified motivational Go/NoGo task capturing Pavlovian choice and instrumental learning bias [23,31]; (2) a probabilistic reversal learning task measuring feedback-based cognitive flexibility [16] and a modified sequential ("two-step") decision-making task to assess model-based control [8,32,37].

### Specific age-dependent developmental changes

We employed a recent variant of a motivational Go/NoGo task designed to tease apart a non-selective Pavlovian influence (i.e., behavioral activation to reward as compared to inhibition to punishment) from selective instrumental influence (i.e., repetition of selective actions after reward and a selective avoidance after punishment [23,31] on choice behavior. For an in-depth task description, refer to the methods and supplement information [SI] (S1 Text and S2 Fig). Using mixed-effects models, we replicated an age-dependent increase of Pavlovian biases from adolescence into adulthood [17]: age predicted more activation (Go responses) to reward cues and more inhibition (NoGo responses) to punishment cues (age × valence: ß = 0.107, SE = 0.05, $\chi^2(1)$ = 5.25, $p$ = 0.02) (Fig 1D). This was accompanied by overall better performance with higher age ($r_s(91)$ = 0.24, $p$ = 0.021, overall task accuracy: mean [SD] = 66.24 [0.18], median [SD] = 69.69 [0.18]). Meanwhile, the selective impact of instrumental learning biases on choices did not relate to age (age × action taken × outcome valence × outcome salience: ß = 0.033, SE = 0.03, $\chi^2(1)$ = 1.73, $p$ = 0.2) (Fig 1E) (see S1 Text and S1–S5 Tables for more details, model statistics and control analyses regarding gender, which did not influence results).

### Computational modeling

As in previous publications, all models included a learning rate. We iteratively added a Gobias parameter (capturing individual's general tendency to make a Go response) and a Pavlovian bias parameter (capturing effects of reward versus punishment cues on activation Go versus inhibition NoGo). All models included either 1 noise parameter across all trials or 2 independent noise parameters for win versus punishment context. Partially in accordance with our previous work [16], we added a computational model with 2 separate feedback sensitivity parameters capturing the degree of noise (i.e., stochasticity) based on positive or negative outcome valence ($\rho_{+FB}$ and $\rho_{-FB}$). Of note, feedback sensitivities are very similar to the inverse temperature parameter known from other RL model implementations [38–40], as they determine how deterministically or stochastically choices follow from learnt action values. They also determine how closely an agent follows a win-stay, lose-shift strategy. Also, higher feedback sensitivity can be interpreted as less decision noise and vice versa. The outcome-based

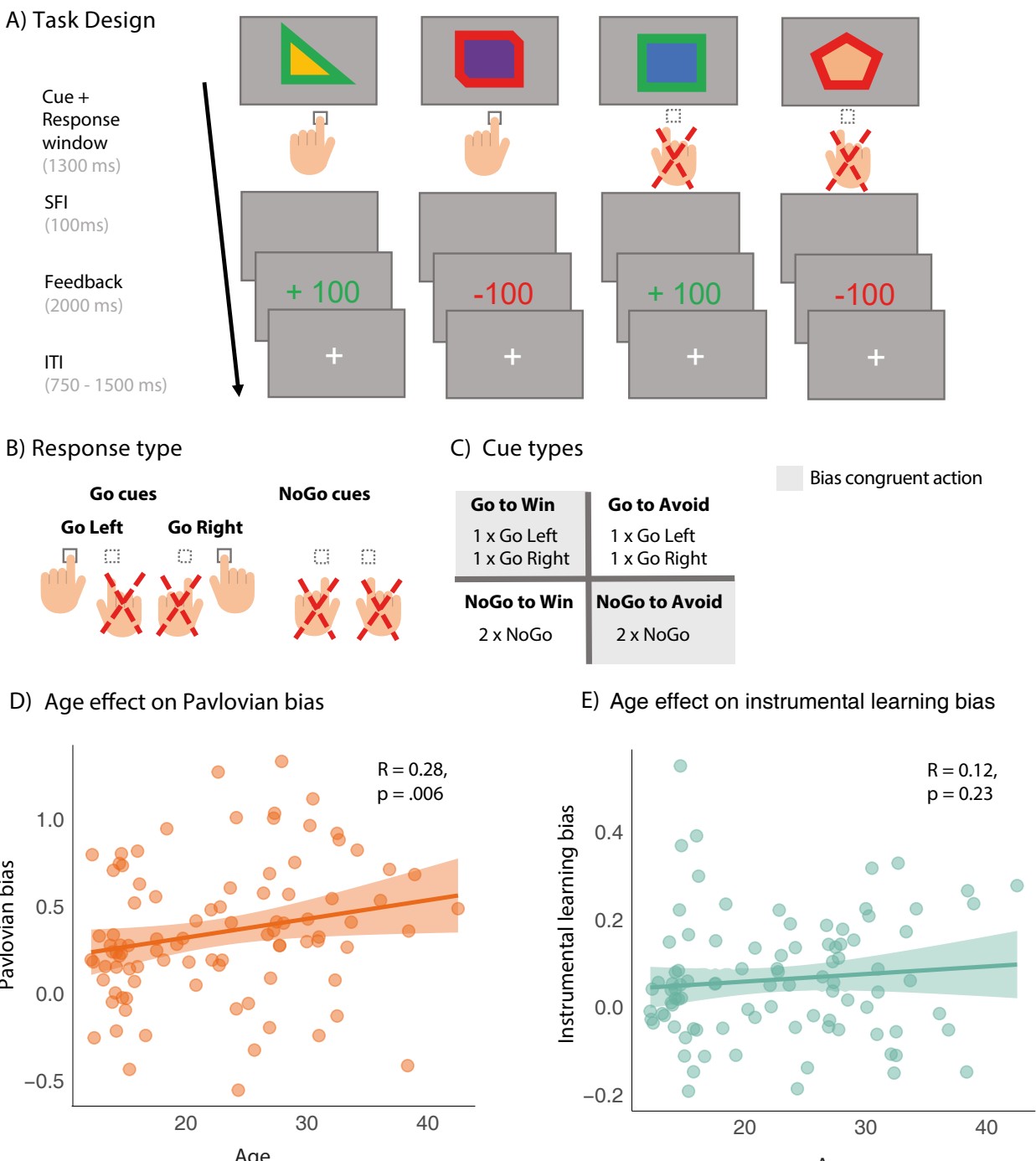

**Fig 1. Task set-up motivational Go/NoGo task.** **(A) Task trial sequence** displayed for all 4 cue categories, Go to Win, Go to Avoid, NoGo to Win, and NoGo to Avoid, but not split up for different Go responses (Go left and Go right). Go left/right responses for Win cues and NoGo responses for Avoid cues are considered bias-congruent, as the cue's respective action requirement matches with the stimulus-response coupling facilitated by the motivational bias. Meanwhile, Go responses for Avoid cues and NoGo for Win cues are considered bias-incongruent response-stimulus couplings. Each cue was presented for 1,300 milliseconds (ms) and participants had to decide whether to execute a Go Left or Go Right response by pressing the respective button or selecting a NoGo response by withholding responding. Subsequently, participants were shown a valid or invalid outcome (reward, neutral, punishment) for 2,000 ms based on the probabilistic feedback schedule (80:20% ratio) and cue valence. The inter-trial-interval (ITI) was 750–1,500 ms, in steps of 250 ms. **(B) Response type.** For the cues with a go response requirement, cues either required a left or a right button press. For the NoGo cues there was naturally no distinction. **(C) Cue types.** Depiction of which cues can be considered bias-congruent (gray shaded box), while the other cues (white box) are cues with bias-incongruent response requirements. **(D) Pavlovian Bias × Age.** Depiction of the

correlation between the individual slope for the valence term extracted from the mixed effects model capturing Pavlovian biases and age. The association shows a clear developmental change of Pavlovian biases across age. **(E) Instrumental Learning Bias × Age**. Depiction of the correlation (spearman) between the individual slope for the interaction term (taken action × outcome salience × outcome valence) extracted from the second mixed effects model capturing instrumental learning biases and age. Here, no evidence for age-dependent changes of instrumental learning biases with age is discernible. Data for figure panels D and E can be found at https://osf.io/mcx36/ together with code for reproducing those parts of the figure.

implementation of feedback sensitivity was indeed superior to the other tested models according to Bayesian model selection (protected exceedance probability (PXP) = 1; model expressed by 95.63%, see Table 1 for model comparison statistics and parameter estimates). Importantly, this model could also predict key characteristic behavioral patterns of the observed empirical data and parameter recovery was excellent (S6 Fig). In line with the mixed models on raw choice data, there was a positive correlation between age and Pavlovian biases ($r_s(91) = 0.23$, $p = 0.024$) and between age and feedback sensitivity to positive outcomes ($r_s(91) = 0.24$, $p = 0.018$), thus replicating previous reports of elevated noise in adolescents [4,16]. Other parameters did not correlate with age ($p > 0.2$) (Figs 2 and S4).

## Noise generalizes across RL tasks

Next, we examined the cross-task generalizability versus context-specificity of noise parameters across several RL tasks: the Go/NoGo task, a probabilistic reversal learning task [16], and the two-step task [8]. For the reversal task, the winning model, as described in [16], comprised 4 feedback sensitivity parameters accounting for effects of motivational context (win reward versus avoid punishment) and feedback valence (positive /+FB versus negative/-FB). For simplicity, we averaged parameter estimates across motivational context, resulting in 2 estimates based on feedback valence. While $\rho_{\text{Go/NoGo +FB}}$ from the Go/NoGo task was positively associated with the reversal feedback sensitivity parameter for positive outcomes $\rho_{\text{Reversal +FB}}$ [$r_s(87) = 0.47$, $p < 0.001$], the reversal feedback sensitivity parameter

**Table 1. Overview of transformed single hierarchical model parameter estimates (mean/standard error) and full (hierarchical) model comparison metrics and model frequencies.** Protected exceedance probability and model frequency from the full model comparison identified model M7 as winning model. PXP = protected exceedance probability. The data and statistics presented in this table can be computed using code provided at https://osf.io/mcx36/.

| | M1 | M2 | M3 | M4 | M5 | M6 | M7 |
|---|---|---|---|---|---|---|---|
| $\rho$ | 25.73 [1.59] | 24.17 [1.52] | 15.37 [1.07] | 21.10 [1.44] | 15.71 [1.01] | | |
| $\rho_{\text{win}}$ | | | | | | 21.06 [1.22] | |
| $\rho_{\text{avoid}}$ | | | | | | 12.60 [0.75] | |
| $\rho_{\text{+FB}}$ | | | | | | | 5.52 [0.24] |
| $\rho_{\text{-FB}}$ | | | | | | | 1.03 [0.10] |
| $\varepsilon$ | 0.04 [0.01] | 0.05 [0.01] | 0.08 [0.01] | 0.06 [0.01] | 0.06 [0.01] | 0.06 [0.01] | 0.18 [0.01] |
| b | | −0.02 [0.06] | 0.07 [0.06] | −0.001 [0.06] | 0.06 [0.06] | 0.02 [0.06] | −0.08 [0.03] |
| $\pi$ | | | 0.34 [0.09] | | 0.06 [0.07] | 0.72 [0.08] | 0.25 [0.03] |
| $\kappa$ | | | | 2.61 [0.33] | 1.68 [0.11] | | |
| **Full model comparison (M1 –M7)** | | | | | | | |
| Model frequency (%) | 0 | 0 | 4.37 | 0 | 0 | 0 | 96.63 |
| PXP | 0 | 0 | 0 | 0 | 0 | 0 | 1 |

Age effects on computational model parameters (winning model M7)

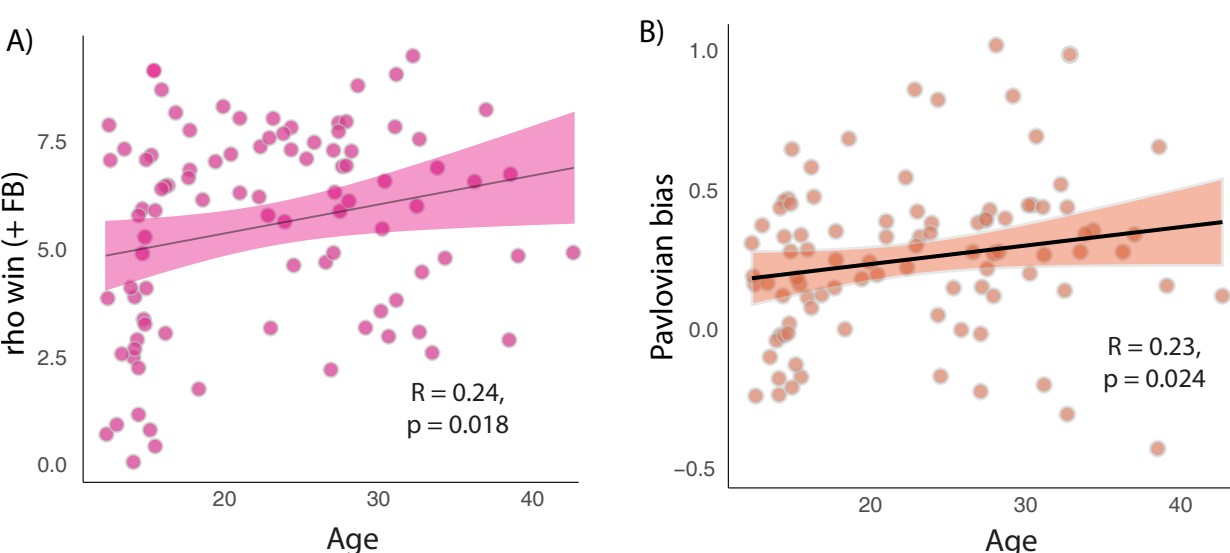

**Fig 2. Significant age effects on model parameters extracted from the winning model M7.** (A) The feedback sensitivity parameter for positive outcomes showed a significant age-dependency, such that older individuals were more sensitive to positive outcomes, as depicted in the scatterplots. (B) The Pavlovian bias parameter showed a significant age-dependency, such that older individuals' decisions were impacted more by motivational biases as captured by the Pavlovian bias parameter. Data underlying this figure and the code for reproducing it can be found at https://osf.io/mcx36/
.

for negative outcomes correlated negatively with it [$\rho_{\text{Reversal -FB}}$: $r_s(87) = -0.3$, $p = 0.004$] (Fig 3A). Both survived multiple comparison correction with $p$-value $< 0.0125$. No correlation between reversal task noise and feedback sensitivity for negative outcomes ($\rho_{\text{Go/NoGo -FB}}$) reached significance ($p > 0.1$).

For the two-step task, both noise parameters, ß1 and ß2, were significantly correlated with Go/NoGo decision noise for positive outcomes (Fig 3B). More decision noise in the Go/NoGo task was associated with more decision noise from the two-step task (ß1: $r_s(90) = 0.43$, $p$-value $< 0.001$; ß2: $r_s(90) = 0.52$, $p$-value $< 0.001$). The association between second stage

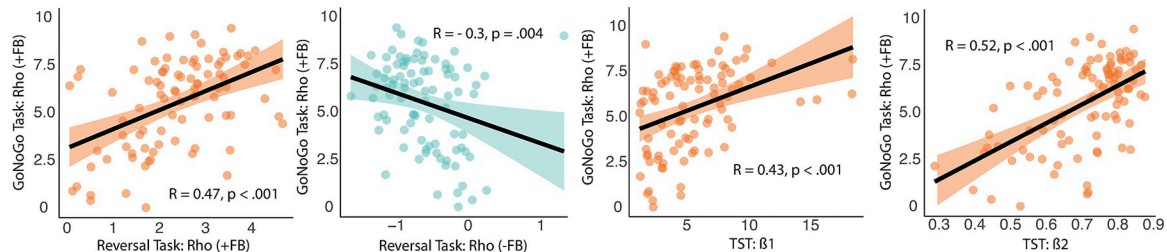

**Fig 3.** (A) Depiction of the association between feedback sensitivity for positive outcomes from the Go/NoGo Task and noise parameters from the reversal task. As expected, all noise parameters which captured noise related to positive outcomes were positively correlated, while negative correlations could be observed when correlating feedback sensitivity for positive outcomes with feedback sensitivity parameters from the reversal task capturing feedback sensitivity for negative outcome. (B) Significant cross-task correlation between both computationally derived noise parameters ß1 and ß2 from the two-step task with the feedback sensitivity parameter for positive outcomes from the Go/NoGo Task. Results were considered significant with $p < 0.05/4$ (0.0125), respectively, thereby correcting for multiple testing. The data underlying this figure and the code for reproducing it can be found at https://osf.io/mcx36/.

decision noise, ß2, and Go/NoGo decision noise for negative outcomes was weaker and barely survived multiple comparison correction ($r_s(90) = -0.26$, $p$-value = 0.012). Cross-task correlation for learning rates from each task proved nonsignificant ($\varepsilon_{Go/NoGo} \times \varepsilon_{Reversal}$: $r_s(87) = --0.027$, $p = 0.8$; $\varepsilon_{Go/NoGo} \times \varepsilon_{TST\ alpha1}$: $r_s(90) = -0.008$, $p = 0.94$; $\varepsilon_{Go/NoGo} \times \varepsilon_{TST\ alpha2}$: $r_s(90) = -0.04$, $p = 0.74$), highlighting the task- or context-specificity of those parameters.

Results of additional correlational analyses of unspecific noise from the motivational Go/NoGo task with general and more specific task performance indices across all 3 tasks are reported in the SI (S1 Text and S8 Fig). In short, less decision noise for positive outcomes in the Go/NoGo task was associated with increased overall accuracy on the Go/NoGo and reversal task (pre-post reversal accuracy) as well as more goal-directed behavior and less switching after negative outcomes in the reversal task. Meanwhile, less decision noise for negative outcomes in the Go/NoGo task was associated with decreased task performance in the GoNoGo task and the reversal task.

## Noise mediates developmental changes in decision processes

A key interest of our study was to examine whether decision noise mediates specific developmental changes in decision processes, critically, across tasks. We assessed mediation effects of noise from the motivational Go/NoGo task (feedback sensitivity for positive feedback) on the association between age and (1) general task performance and non-selective Pavlovian Bias on the Go/NoGo task; (2) performance and loose shift behavior on the reversal task; and (3) model-based behavior on the 2-step task. Given that feedback sensitivity for negative feedback showed no association with age, this was done specifically for feedback sensitivity for positive feedback.

### Within-task mediation: Motivational Go/NoGo task

We set up a mediation model with the age-dependent noise parameter (sensitivity to positive feedback) from the computational model of the motivational Go/NoGo task as mediator of the relationship between age and general performance in the task and Pavlovian bias. We found a significant partial mediation effect for $\rho_{+FB}$ ($p = 0.02$), accounting for 83,6% of the total effect of the relationship between age and overall performance (Indirect effect = 0.004, CI [0.0005–0.01], p = 0.03; direct effect = 0.0009, CI [−0.001–0.003], $p = 0.5$; Total effect = 0.005, CI [0.0008–0.01], $p = 0.02$). Next, we examined whether the association between age and the score computed for overall Pavlovian bias [$P_{corr}$(Go2Win)—$P_{corr}$(Go2Avoid) + $P_{corr}$(NoGo2Win)—$P_{corr}$(NoGo2Avoid)] or the computationally derived parameter capturing the Pavlovian bias was mediated by positive feedback sensitivity. This association between age and Pavlovian bias was not significantly mediated by positive feedback sensitivity, neither for the task score ($p$-value = 0.3) nor for the computational parameter capturing the effect of Pavlovian biases ($p$-value = 0.5).

### Cross-task mediation: Reversal task

Two mediation models were computed to determine the mediating effect of feedback sensitivity for positive outcomes (noise) in the Go/NoGo task on performance parameters derived from the reversal learning task, namely pre- minus post-reversal accuracy and switching following negative outcomes, both of which have previously been shown to correlate with age [16]. Assessment of the mediation effect of noise on the association between age and pre- minus post-reversal accuracy provided evidence for a partial mediation with $\rho_{+FB}$ accounting for up to 37.1% of the total effect ($p = 0.03$) (Indirect effect = 0.0016, CI [0.0002–0.004], $p = 0.02$; direct effect = 0.003, CI [−0.0004–0.01], $p = 0.08$; Total effect = 0.005, CI [0.001–

## A) Reversal Task Design

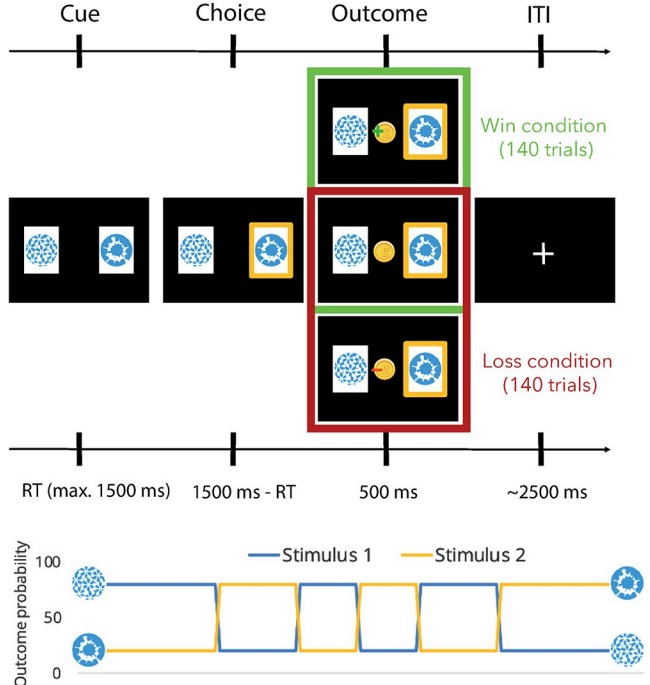

## B) Modified sequential (2-step) learning task

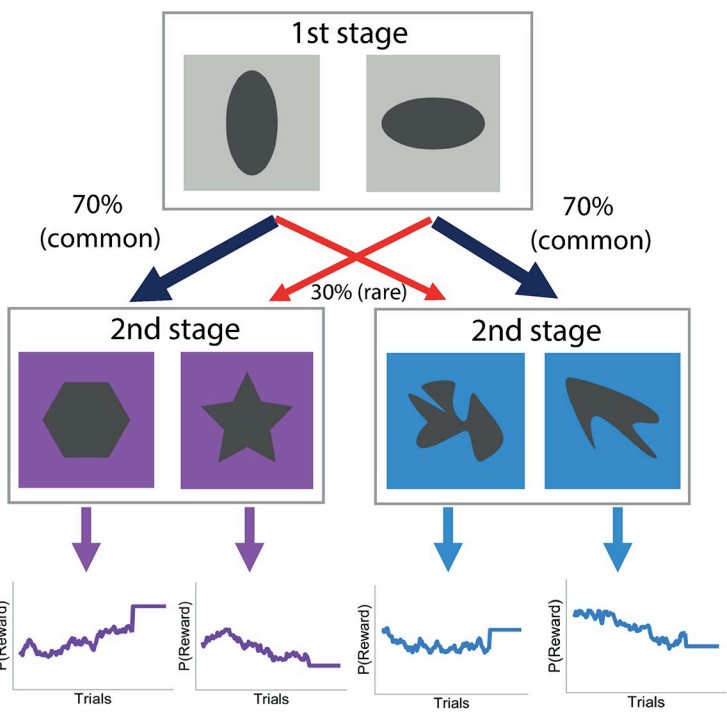

## D) Mediation effects of noise on reversal

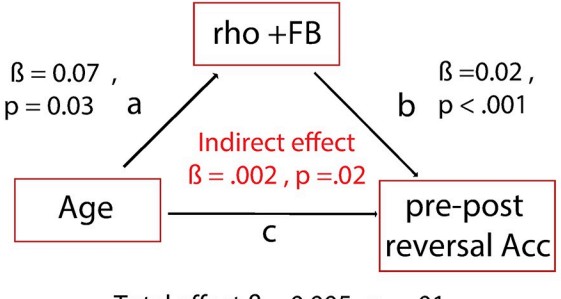

Total effect ß = 0.005, p = .01

## D) Mediation effects of noise on MB control

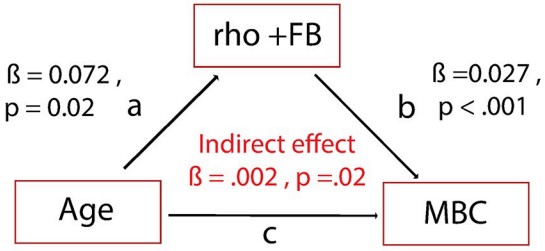

Total effect ß = 0.005, p = .004

**Fig 4. (A) Probabilistic reversal task design.** In this task, participants had to adapt their behavior according to the changing outcome probabilities across the reversal learning task. Image of task design modified from Waltmann and colleagues [16]. **(B) Task design of sequential decision-making task.** In this 2-step task, a choice on the first stage led to one of 2 possible second stages. Here, participants had to make a second choice, upon which participants received a reward or neutral outcome (rewards were replaced by punishments in the punishment context). The probability of receiving a reward/punishment was determined by constantly changing probabilities, i.e., based on Gaussian random walks, while transition probabilities to transfer from stage 1 to stage 2 were fixed. They were either considered common transitions (70%) or rare (30%). Image modified from Scholz and colleagues [8]. (C) Mediation analysis reversal task indicating a significant mediation effect of noise from positive outcomes on the association between age and pre minus post reversal accuracy. (D) **Mediation analysis** 2-step task showing the significant mediation effect of feedback sensitivity for positive outcomes on the relationship between age and model-based control. *P*-values below 0.025 were considered significant thereby correcting for multiple testing.

0.01], $p = 0.01$ (Fig 4C). In the second mediation model, $\rho_{+FB}$ accounted for up to 26,03% of the total effect between age and switching after negative feedback (p = 0.02) (Indirect effect = −0.002, CI [−0.003 −−0.0002], $p = 0.02$; direct effect = 0.005, CI [−0.007–0.001], $p = 0.005$; Total effect = −0.006, CI [−0.009 −−0.003], $p < 0.001$) (Fig 4A).

## Cross-task mediation: 2-step task

Assessment of the mediation effect of feedback sensitivity for positive outcomes in the Go/NoGo task on the association between age and model-based control provided evidence of a partial mediation with $\rho_{+FB}$ accounting for up to 44,01% of the total effect of the relationship of age and model-based control ($p$ = 0.02) (Indirect effect = 0.002, CI [0.0004–0.004], $p$ = 0.02; direct effect = 0.003, CI [−0.0003–0.01], $p$ = 0.08; Total effect = 0.005, CI [0.002–0.01], $p$ = 0.004) (Fig 4D). Rerunning the same mediation model using the computationally derived parameter omega (weight parameter capturing the balance between model-free and model-based control) indicated a similar, though somewhat weaker mediation effect (19.75%, $p$-value = 0.017) (see Fig 4B).

## Discussion

In this study, we relied on computational modeling across 3 distinct RL tasks to assess a novel mediating role of decision noise—known to be elevated in adolescents—on age-dependent increases in more sophisticated choice behaviors and performance gain. This mediating role referred to model-based control, switching after negative outcomes on the reversal task and performance gain in both the Go/NoGo and reversal task. A choice heuristic like the Pavlovian bias was not mediated by decision noise. In line with this mediating role of decision noise, we also confirm previous findings indicating decision noise as a rather stable characteristic across tasks and contexts.

Using computational modeling, we assessed decision noise as latent feature of the decision process during development. We show an age-dependent increase of feedback sensitivity specifically for positive outcomes, as did [16] in a different task from the same sample. This is in line with well-known decreases of noise with higher age [27,41] and previous reports of more random "noisy" choice behavior in adolescents [4,16,27]. Consistent with previous work [4,26,42], we show stable cross-task generalization of decision noise by means of strong correlations of noise parameters across tasks. This reinforces the notion of decision noise as a task-independent feature with substantial interindividual differences. In contrast, cross-task correlations of learning rates indicated a lack of generalizability, much in accord with previous studies showing task-specificity of learning rates [4,26]. One tentative explanation for this is that learning per se, unlike decision noise, may be highly context dependent [26]. This might explain why specific developmental effects appear more inconsistent across (RL) tasks: while Rosenbaum and colleagues (2022) reported specific developmental effects on punishment learning rates in the absence of effects on reward learning rates, Pauli and colleagues (2023) described elevated reward learning rates [43]. Thus, even subtle difference in a task's design may considerably impact the way adolescents learn and thus result in the detection of distinct effects, while individual differences in noise seem more robust against such differences in task design [4].

Critically, the central novel finding of our study indicated a mediating effect of this seemingly "unspecific" but stable noise on age-dependent increases in specific and more sophisticated choice signature across different task settings. Here, mediation analyses revealed that a substantial part of the variances between age and MB control, switching behavior after negative outcomes as well as overall task performance on the reversal and Go/NoGo task were accounted for by age-related developmental differences in noise levels. Thus, "unspecific" noise mediates the development of highly specific functions or strategies.

One reason for these mediation effects could be a limited availability of cognitive resources in adolescents [44–47] due to the ongoing development of brain areas related to cognitive control [1,48,49]. Having fewer cognitive resources might make adolescents more prone to rely on

computationally cheaper decision strategies, rendering them more susceptible to emotional, motivational, and social influences [28,50,51]. Still, other work has shown adolescents to employ more complex strategies relative to adults when mentalizing and processing social emotions [52]. The exertion of cognitive control as value-based choice, i.e., the willingness to allocate and exert control in certain situations [53–58] might be another possible explanation. Here, cognitive control is a choice, governed by a cost-benefit tradeoff, where people choose to exert control whenever this will result in a large enough increase in expected reward. Individuals can thus learn to selectively exert control, when this returns additional reward, but refrain from it if costs outweigh the benefits [59,60]. More noisy choice behavior, i.e., rather random-like exploration, may thus constitute a somewhat "rational" choice by adolescents to not mobilize control to reduce effort expenditure while achieving higher levels of control, may seem too costly. Alternatively or in addition, Ma and colleagues (2022) suggested that elevated explorative (or noisier) behavior might serve the development of central social behavior in adolescence [61], such that choice uncertainty (comparable to decision noise) predicts contagion effects of peer's choices, which may be beneficial for social integration [62].

Of note, decision noise can reflect distinct underlying processes, such as random or more sophisticated directed exploration [30]. For example, directed exploration leads individuals to occasionally stray from selecting the optimal choice to deliberately choose less known options to gather information [44,45] to maximize long-term outcomes. Meanwhile, random exploration refers to randomly choosing options, a pattern resulting in more frequent choices of the non-optimal option, rather than deliberately choosing the worse option [45]. According to Findling and colleagues (2019) [45], the majority of choices that seemingly do not optimize reward values, appear to originate from so called learning noise. The locus-coeruleus-norepinephrine system has been proposed as potential neural correlate underlying learning noise [44,45] and using psychopharmacological manipulation, has been implicated in computationally "cheaper," value-free exploration strategies [44,45]. Meanwhile, more elaborate exploration strategies appear to rely on other systems, like the dopaminergic system [63,64]. Future studies may disentangle such distinct noise components and its neurochemical correlates more precisely.

Interestingly, in our data no age effect was evident for decision noise for negative outcomes. Speculatively, this could be linked to relatively stronger effects of development on the reward versus punishment domain [8,16,43]. Alternatively, the parameter for decision noise for negative outcomes might capture different aspects of noise that are less affected by developmental changes, though we cannot make any more specific claims about this, as our Go/NoGo task cannot dissect distinct subcomponents of noise such as decision from learning noise.

One avenue of future research is distinct patterns of decision noise in developmental psychopathology. Indeed, elevated decision noise was reported across a wide range of psychiatric disorders [65–69]. Developmentally, a particular interesting condition associated with noisy behavior is attention-deficit hyperactivity disorder (ADHD). Some reports show elevated noise and exploration in ADHD patients [69–71] and non-clinical samples reporting ADHD symptoms [72,73]. Further assessment of the computational phenotype of ADHD leading to the characteristic profile of undirected explorative behavior, rapid task switching, and inattention might be key [30,74]. Such assessment would be particularly exciting given the implication of dopaminergic and noradrenergic pathways not only in decision noise and the exploration/exploitation trade-off [29,63,64,75] but also in the psychostimulant treatment of ADHD symptoms [76–81]. It remains unknown whether decision noise could serve as predictor for clinical outcomes like individual differences in response to psychostimulant treatment.

With respect to Pavlovian biases, we replicate a reduction during adolescence [17,20] but alongside increasing task performance from adolescence into adulthood. The latter finding on

performance conflicts with previous work, where lower Pavlovian biases presented with improved task performance, especially when participants had to actively "overcome" inherent Pavlovian responding [17]. This discrepancy might be explained by different task versions such as different number of Go responses, outcome types, and cued valence. We extend previous work [17,20] by showing that instrumental learning biases in this type of task do not undergo significant developmental changes.

As for limitations, a sample including younger children would have been superior by making our sample more comparable to previous work [17]. As data was collected as part of a larger study, this was not feasible. Still, we were able to partially replicate findings by showing a linear trend based on the particular age range available to us. While our study had a cross-sectional between-subject design, a longitudinal, within-subject design with multiple measurement points could have better captured developmental changes in the examined RL processes.

In sum, using computational modeling and mediation analysis, we showed that decision noise had a significant mediating effect on age-dependent increases in higher-level cognitive processes such as model-based control, switching after negative outcomes in the reversal task and overall performance in the motivational Go/NoGo and the reversal task. Cross-task analyses also emphasized decision noise as representing an interindividually more stable parameter, maybe even a trait-like feature. Future work may unravel the neural basis as well as the developmental and clinical real-life relevance of decision noise for neurodevelopmental disorders such as ADHD to bridge the gap between observed symptom-level behavior and neurobiological mechanisms. Moreover, given that many of the cognitive processes we measure appear to be at least in part impacted by noise, future studies should attempt to quantify the degree of noise relative to the central cognitive processes under investigation.

## Material and methods

### Sample

We recruited 103 participants as part of a larger developmental study, all of whom were screened for current psychiatric diagnosis. Participants completed several RL tasks, such as a reversal learning task capturing behavioral flexibility [16], a 2-step task measuring model-based control [8] and a well-established motivational Go/NoGo task assessing motivational biases in decision-making [23,31] (for a more detailed study description refer to the preregistered study protocol at https://doi.org/10.17605/OSF.IO/FYN6Q [36]). Of those 103, 99 participants completed the Go/NoGo task, of which 93 were subsequently analyzed ($n$ = 93: age mean [SD] = 22.65 [7.88], age range 12 to 42 years, female: 45; male: 48), as $n$ = 5 participants did not meet a rudimentary performance check (see SI) and one was an age outlier; 40% ($n$ = 37) of our final sample were adolescents (18 years of age or below). The age distribution was not uniform but right-skewed (S1 Fig), as participants had been initially recruited as matched controls for a clinical sample in terms of age and gender. Participant reimbursement was 9 Euro per hour for study participation. Study proceedings were in agreement with the declaration of Helsinki and approved by the ethics board of the medical faculty at the University of Leipzig (385/17-ek). All participants were informed about the study proceedings and provided informed written consent before participating in the study.

### Motivational Go/NoGo task

For this study, we used a well-established probabilistic reinforcement learning task known to experimentally measure motivational biases by examining the impact of valence (gain win versus avoid punishment contexts signaled by cues) on behavioral activation or inhibition (Go versus NoGo action) [23,31]. Here, on each trial, study participants were shown a cue for

which they had to decide whether to execute one of 2 Go responses (make a right or left button press) or abstain from it (NoGo, no button press) to either win a reward (Win cues) or avoid a punishment (Avoid cues) (Fig 1). Importantly, participants were aware whether they were playing for rewards or avoiding punishments, as cue valence was cued using a colored frame around each cue (green = Win cue; red = Avoid Cue). However, despite showing the correct response for a cue, participants could still receive invalid feedback 20% of the time (versus 80% valid feedback) according to probabilistic feedback. Participants could show a correct left Go response for a Win cue requiring a left Go response and still receive neutral feedback, the non-favorable outcome for a Win cue, on this trial. They heart a specific sound for receiving a reward, neutral feedback, or punishment. The task comprised 320 trials in total and each one of the 8 cues was presented 40 times. Before starting the main task, participants completed practice trials to guarantee that participants had understood task requirements such as the possibility of a NoGo response.

The impact of choice biases is operationalized by how well participants learn to show or omit a response when facing a reward or punishment cue requiring a Go or NoGo choice as optimal response. Consecutive choices represent learning of optimal choices and are guided by probabilistic feedback (rewards/neutral feedback for reward cues; punishment/neutral feedback for punishment cues). Importantly, bias-congruent responding (Go response for a Go-to-Win cue; NoGo for Avoid Punishment cue) should be facilitated, i.e., participants show better performance on these trials, while bias-incongruent performance should be impeded.

To tease apart differences in Pavlovian choice from instrumental learning biases, this task version had 2 Go responses, Go Left and Right [23,31,82]. This manipulation discerns whether, for example, a previously rewarded optimal Go response (e.g., Go Left) will be specifically reinforced and repeated or omitted more frequently in subsequent trials with the same cue [23,31]. This specificity for the optimal (Go) response reflects an instrumental learning bias, which will increase throughout the task while participants learn the optimal cue response, while the impact of Pavlovian choice bias recedes across the task [23]. Overall, instrumental learning biases are more selective when compared with Pavlovian biases.

Unlike the instrumental learning bias, the Pavlovian bias does not distinguish between the type of Go response, here Go Left or Go Right, in its facilitation effect, so any Go response followed by a reward will increase the likelihood that a Go response will be selected on the next trial this cue is presented. Also, Pavlovian biases usually do so as early as the first trial [23]. They are also characterized by a nonspecific tendency to show more Go responses for Win versus Avoid Punishment cues.

## Mixed model analysis of Go/NoGo task

To examine Pavlovian choice biases, we assessed whether the probability of making a Go response, subsequently termed P(Go), was impacted by the within-subject factors required action (Go versus NoGo) and valence (Win versus Avoid Punishment) while also including age (z-standardized) as additional covariate of interest. We expected a linear age effect for Pavlovian biases based on previous reports, such that adolescents would display lower levels of Pavlovian biases relative to adults [17,20]. We were therefore particularly interested in the 2-way interaction valence × age indicating whether Pavlovian biases change with age, alongside the main effects of (1) required action representing whether individuals actually learn to make the correct response; (2) the main effect of valence capturing the presence of a motivational bias on choice behavior.

P(Go) ~ required action* cue valence*age + (required action* cue valence + 1|Subject)

To examine developmental effects on instrumental learning biases, we set up a second model based on previous work [31]. Here, we tested whether the probability of repeating the previous response P(repeat), changed depending on 3 within-subject factors, namely the response selected on the previous trial (Go versus NoGo response), the outcome valence (positive: reward for reward cues; neutral feedback for avoidance cues versus negative: punishment for avoidance cues; neutral feedback for reward cues) and outcome salience (salient: reward/punishment feedback, non-salient = neutral feedback). Importantly, in this model, the presence of an instrumental learning bias is indicated by a significant 3-way interaction: action taken × outcome valence × outcome salience. As we were specifically interested in the presence of age-dependent effects, the model also included a linear age term (z-standardized) as covariate of interest (also see S1 Text for an alternative model implementation). Prior behavioral and neural work [83–85] has reported age-dependent differences in feedback-based, instrumental learning processes, for instance heightened negative (relative to positive) learning rates during adolescence [85]. Consequently, we expected to observe reduced selective response facilitation of the correct response for a reward cue during adolescence.

P(repeat) ~ action taken $_{t-1}$ * outcome valence $_{t-1}$ * outcome salience $_{t-1}$ *age + (action taken $_{t-1}$ * outcome valence $_{t-1}$ * outcome salience $_{t-1}$ + 1|Subject)

Given previous evidence of differences in age of onset of puberty [86], we also ran models including gender as additional control variable, thereby assessing potential effects of gender on choice behavior and biases.

All generalized logistic mixed effects models were computed using the lme4 package, version 1.1–31 in R 4.2.2 with the optimizer bobyqa and the maximal number of iterations set to $n = 1e+9$. Statistical significance was determined using $p$-values with $α < 0.05$, two-sided.

## Computational modeling of Go/NoGo task and age-dependent changes

To dissect the computational mechanisms sub-serving Pavlovian action biases and instrumental learning, we fitted several hierarchically nested RL (M1-M7). For this, we relied on the cbm toolbox implemented in matlab [87], which is based on hierarchical Bayesian inference (HBI) and treats the model itself as a random effect [88,89]. Models M1–M5 have been previously employed and outlined in much detail by Swart and colleagues [23]. In brief, model M1 represented a Rescorla Wagner model [90] comprising a learning rate ($ε$) and a second parameter capturing feedback sensitivity, to learn value of each respective action (a: Go left, Go right, NoGo) for each stimulus (s) on each trial t:

$$Q_t(a_t, s_t) = Q_{t-1}(a_t, s_t) + ε(ρr_t - Q_{t-1}(a_t, s_t)) \tag{1}$$

M2 is an extension of M1 with an additional "Gobias" parameter $b$ capturing an overall tendency to give a Go response. Model M3 extended model M2 with another parameter operationalizing the Pavlovian tendency $π$ to show more Go responses for Win relative to Avoid Punishment cues. Both bias parameters were integrated with the learnt Q values into the action weights $w$:

$$w_t(a_t, s_t) = \begin{cases} Q_t(a_t, s_t) + b + Vπ(s) \ if \ a = Go \\ Q_t(a_t, s_t) \ else \end{cases} \tag{2}$$

Model M4 included an instrumental learning bias parameter $\kappa$ instead of the Pavlovian parameter $\pi$ to assess whether the choice behavior could have been solely produced by a learning bias. For this, $\kappa$ was included as a modification of the learning as follows:

$$\varepsilon = \begin{cases} \varepsilon_o + \kappa \; \textit{if } r_t = 1 \textit{ and } a = \textit{Go} \\ \varepsilon_o - \kappa \; \textit{if } r_t = -1 \textit{ and } a = \textit{NoGo} \\ \varepsilon_o \; \textit{else} \end{cases} \tag{3}$$

Importantly, to ensure a symmetric impact of $\kappa$, the following requirements were implemented depending on the size of the learning rate

$$\varepsilon = \begin{cases} \varepsilon_o = \textit{inv.logit}(\varepsilon) \\ \varepsilon_{\textit{punished NoGo}} = \textit{inv.logit}(\varepsilon - \kappa) \textit{ if } \varepsilon_o < .5 \\ \varepsilon_{\textit{rewarded Go}} = \varepsilon_o + (\varepsilon_o - \varepsilon_{\textit{punished NoGo}}) \textit{ if } \varepsilon_o < .5 \\ \varepsilon_{\textit{rewarded Go}} = \textit{inv.logit}(\varepsilon + \kappa) \textit{ if } \varepsilon_o > .5 \\ \varepsilon_{\textit{punished NoGo}} = \varepsilon_o + (\varepsilon_o - \varepsilon_{\textit{rewarded Go}}) \textit{ if } \varepsilon_o > .5 \end{cases} \tag{4}$$

For all models, V denoted the cued valence ($V_{\text{win}} = + 0.5$; $V_{\text{avoid}} = -0.5$). Consequently, a positive $\pi$ lead to an increased action weight for Go responses for Win cues, while resulting in a reduced action weight for Go responses on Avoid cues. Action weights were transformed to action probabilities using a softmax function:

$$p(a_t|s_t) = \frac{\exp(w(a_t, s_t))}{\sum_a \exp(w(a', s_t))} \tag{5}$$

Model M5 included both bias parameters $\kappa$ *and* $\pi$. Due to previous reports of distinct learning and processing of positive and negative feedback in adolescents [8,16], we aimed to examine whether feedback sensitivity parameters for each motivational context ($\rho_{\text{win}}$ and $\rho_{\text{avoid}}$) or feedback sensitivity parameters for positive versus negative feedback ($\rho_{+\text{FB}}$ or $\rho_{-\text{FB}}$; present in win and avoid motivational context) would provide a better account of the data (see Table 1 for an overview of the model space). This was motivated by rather noisy behavior seen in adolescents in similar RL tasks [4,16,27]. Information of parameter transformation can be found in the SI.

We performed extensive model simulation based on the established models to rigorously compare the observed relative to the simulated data (S5 Fig). This revealed that the model only including an instrumental learning bias parameter $\kappa$ captured the observed behavioral data very poorly. In the same veine, model M5 including both motivational bias parameters, $\pi$ and $\kappa$, did not perform better compared to model M3, which only included the Pavlovian bias parameter $\pi$ (see S1 Text, S5 and S6 Figs for details on simulations, model validation, and parameter recovery). Hence, for the purpose of parsimony, we focused on 2 additional model extensions, model M6 and M7, which implemented 2 feedback sensitivity parameters for cue valence (Win and Avoid Punishment) or for positive and negative outcome valence together with a single learning rate, a Gobias and a Pavlovian bias parameter (see S1 Text for full model space).

We concluded this analysis using random effects model comparison [88,89], which computes the Laplace approximation of model evidence based on the individual level [38,91], from which group model evidence is derived to establish which model best captured the behavioral data. Model evidence was examined by comparing the PXP [87]. PXP assesses the most frequently expressed model [88] while accounting for the possibility of chance results. We then

extracted hierarchical model parameter estimates from the winning model and examined age-dependent effects using spearman correlations.

As we were particularly interested in the developmental changes underlying decision noise and biases in choice behavior [16,17,26,92], we evaluated these associations for the respective computational parameters from the winning model with age using spearman correlation coefficients. Given our specific hypotheses regarding the developmental pattern of those three parameters, we considered these confirmatory analyses for which we did not apply multiple comparison correction. For completeness, we also report associations for age with the remaining 2 parameters, namely the learning rate and the Gobias in the supplement.

## Cross task generalizability of unspecific decision noise

Given our primary interest in the cross-task generalizability of decision noise, we subsequently assessed whether feedback sensitivity parameters from the motivational Go/NoGo task would be associated with related parameters from 2 other RL tasks, detailed results for which have been published elsewhere [8,16]. Apart from the motivational Go/NoGo task, we had access to data from a probabilistic reversal learning task capturing cognitive flexibility and a sequential probabilistic decision-making task assessing model-based control (see S1 Text for task details).

We first examined the association between computationally modeled noise parameters across the Go/NoGo task ($\rho_{+FB}$ and $\rho_{-FB}$) and the probabilistic reversal learning task using spearman correlations. For the reversal task, noise parameters were extracted from the winning computational model that had contained 4 noise parameters accounting for both outcome and cue valence, namely, $\rho_{Win +FB}$, $\rho_{Win -FB}$, $\rho_{Loss +FB}$ and $\rho_{Loss -FB}$ [16]. Those estimates were further simplified by averaging parameters on the dimension of cue valence, thereby creating 2 parameter estimates, namely, $\rho_{Reversal +FB}$, $\rho_{Reversal -FB}$. In total, this meant 4 correlations were computed across both tasks. Second, we assessed cross-task associations for noise parameters from the Go/NoGo task ($\rho_{+FB}$ and $\rho_{-FB}$) and the 2-step task, for which noise parameters ß1 and ß2 were extracted from a well-established hybrid model (see [38] for an extensive model description).

## Associations between unspecific noise with specific cognitive functions

Next, we also assessed the relationship between the 2 feedback sensitivity parameters (decision noise) extracted from the winning model for the Go/NoGo task and the index capturing MB control from the modified 2-step RL task (reported in [8]). Given that we did not find significant valence differences for MB control in our previous developmental work [8], we only assessed associations for noise parameters and overall MB control. Furthermore, we computed correlation between the Go/NoGo task noise parameters with the behavioral indices from the reversal learning task, namely pre-post reversal accuracy and switching after negative outcomes.

To determine the specificity of findings, we evaluated the cross-task association between learning rates from the Go/NoGo task, the 2-step and reversal learning task. Here, based on prior work suggesting considerable variation in learning rates based on situational context [4,26], we did not expect cross-task learning rates to show significant correlations.

Results were considered significant with a $p$-value $< 0.0125$ for associations between Go/NoGo and the reversal task parameters (0.05/4) and a $p$-value $< 0.025$ for the associations between the Go/NoGo task and the 2-step task parameters (0.05/2) to correct for multiple testing. This was done separately for each task, as both tasks were considered independently.

## Decision noise as a mediator for higher-level cognitive processes?

To address our second aim for this paper, we also computed several mediation analyses. All included feedback sensitivity as mediator variable and assessed its significance for age related changes on specific cognitive functions or decision processes, namely MB control, decision biases, and markers of cognitive flexibility. The mediation package in R was used for all mediation analysis. Results are reported based on nonparametric bootstrap confidence intervals based on the percentile method (simulations $n$ = 10.000).

## The impact of decision noise on Go/NoGo task performance and Pavlovian biases

We first assessed whether the noise parameter for positive outcomes extracted from our winning computational model M7 ($\rho_{+FB}$) exerted a mediating effect on the relationship between age and overall task performance (P(correct)), on a previously employed score computed for the overall impact of Pavlovian biases on choice behavior [$P_{corr}$(Go2Win)—$P_{corr}$(Go2Avoid) + $P_{corr}$(NoGo2Win)—$P_{corr}$(NoGo2Avoid)] [23] as well as the computational parameter capturing Pavlovian biases.

## The role of decision noise for the association between age and cognitive flexibility

Based on [16], the difference between pre and post reversal accuracy as well as the degree of switching behavior after negative feedback showed significant correlations with age. Hence, we examined whether these parameters were mediated by the noise parameters for positive outcomes from the motivational Go/NoGo task.

## The impact of decision noise on the association between age and MB control

Finally, given previous reports linking age and MB control [8], we also assessed whether this association might be (partially) mediated by the extracted noise parameter for positive outcomes from our winning model M7, namely $\rho_{+FB}$. MB control was operationalized in 2 different ways, namely, the computationally derived parameter omega extracted from a well-established hybrid model [38] as well as effect estimates extracted from a mixed-effects model characterizing MB control (details reported in [8]).

## Supporting information

**S1 Text. Supplemental material.** File providing additional information on data analysis and results including mixed and computational models as well as correlational analysis and the employed tasks.
(PDF)

**S1 Table. Age-dependent changes in Pavlovian biases.** Table displaying the ß estimates, the standard error (SE) as well as statistics for the main and interaction effects from the mixed-effects model computed to assess the impact of age on Pavlovian biases as well as general learning of the task. Here, the dependent variable was the probability of making a go response P (Go). Data and code to compute the statistics presented in this table is available at https://osf.io/mcx36/.
(PDF)

**S2 Table. Age-dependent changes in accurate task performance.** Table displaying the ß estimates, the standard error (SE) as well as statistics from the mixed-effects model computed to assess the impact of age on Pavlovian biases as well as general learning of the task. Here, the dependent variable was the probability of making a correct response P(Correct). Data and code to compute the statistics presented in this table is available at https://osf.io/mcx36/.
(PDF)

**S3 Table. Age-dependent changes in instrumental learning biases.** Table displays the ß estimates, standard errors (SE) as well as statistics from the mixed-effects model computed to assess the impact of age on instrumental learning biases. Here, the dependent variable was the probability of repeating the same response for a given cue P(repeat). Data and code to compute the statistics presented in this table is available at https://osf.io/mcx36/.
(PDF)

**S4 Table. Effects of gender on Pavlovian biases.** Table providing an overview of the ß estimates, standard errors (SE) as well as statistics from the mixed-effects model computed assessing the impact of gender on the measured task effects. Here, the dependent variable was the probability of making a Go response P(Go). Data and code to compute the statistics presented in this table is available at https://osf.io/mcx36/.
(PDF)

**S5 Table. Effects of gender on instrumental learning biases.** Table providing an overview of the ß estimates, standard errors (SE) as well as statistics from the mixed-effects model computed assessing the impact of gender on the instrumental learning bias. Here, the dependent variable was the probability of repeating the same response for a given cue P(repeat). Data and code to compute the statistics presented in this table is available at https://osf.io/mcx36/.
(PDF)

**S1 Fig. Distribution of Age and gender.** (A) Age distribution (age between 12 and 43 years) with the solid line indicating the mean age of 22.65 for this sample ($n$ = 93). (B) Gender distribution of sample. (Female = 45, Male = 48).
(EPS)

**S2 Fig.** (A) Conceptualization of motivational choice biases. **(B) Probability of P(Go) as a function of required action and cue valence**. Learning is apparent from the increased frequency of Go responses for Go cues. The impact of motivational biases is evident from the decreased probability of Go responses for Avoid cues and the increased frequency of Go responses towards Win cues independent of the actually required action. **(C) Probability of P(Repeat) as a function of the action taken and outcome valence.** Here, outcome valence is additionally split up by salience, i.e., whether participants received an actual reward or punishment or neutral feedback. Probability of repeating an action depicted as a function of outcome received on the previous trial and its salience. Cue categories are abbreviated as follows: G2W = Go to Win, G2A = Go to Avoid Punishment, N2W = NoGo to Win, N2A = NoGo to Avoid Punishment. The data underlying the figure panels B and C as well as the code for plotting it can be found at https://osf.io/mcx36/.
(EPS)

**S3 Fig. Depiction of age effects on the interaction action x valence.** (A) Scatterplot depicting the association between age and model estimates for interaction term valence × action extracted from the mixed-effect model with PGo as dependent variable. (B) Scatterplot showing the correlation between age and the approach bias computed based on the raw data PGo (G2W - G2A). (C) Scatterplot showing the correlation between age and the avoid component

of the Pavlovian bias computed based on the raw data PGo (NG2W - NG2A). Data and code necessary to replicate this figure can be found at https://osf.io/mcx36/.
(EPS)

**S4 Fig. Age effects on computational model parameters.** Scatterplots depicting significant age effects on feedback sensitivity for positive outcomes and Pavlovian biases, while showing age independent trajectories for the remaining computational parameters, namely, feedback sensitivity for negative outcomes, the learning rate parameter and the go bias, for which (spearman) correlation coefficients were all non-significant ($p$-value > 0.5). The code with which the data underlying this figure was produced and for plotting this figure can be found at https://osf.io/mcx36/.
(EPS)

**S5 Fig. Posterior predictive model simulations.** (A–C) Panels A–C depict the aggregated results of model simulations computed for models M3-M5 (colored lines) compared to the actually observed data (gray colored lines) to determine whether those models can capture the basic patterns seen for the actual observed behavioral data. Here, simulations ($n = 100$) computed new choices and outcomes according to response probabilities that were based on the optimal parameter estimates generated for the respective model and which were subsequently averaged across all subjects. Here, key elements of the behavioral pattern are for instance whether participants learnt the task (i.e., showed the correct Go and NoGo responses for the respective Go vs. NoGo cues) and whether a strong valence effect characteristic for the influence of Pavlovian biases, such as an increased frequency of Go responses for Win cues relative to Avoid Punishment cues, could be detected. (Panel A–C) Trial by trial estimates of the probability of showing a Go response for models M3–M5. Here, M3 only contains a Pavlovian bias parameter alongside one overall feedback sensitivity parameter, a learning rate and a gobias parameter, M4 exchanged the Pavlovian bias parameter ($\pi$) with a learning bias parameter, while model M5 contained both bias parameters. It becomes evident that only model M3 containing a Pavlovian bias parameter only is able to capture the observed behavioral patterns, while both M4 and M5 show major divergence from the behavioral data indicating strong effects of over- and underestimation of the observed data. (Panel D–F) Depiction of the difference score between the probability of repeating the same choice shown on the previous trial for the same cue on the next trial [P (Repeat)] with the predicted choices based on the simulation runs being subtracted from the originally observed choices and subsequently averaged across participants. Here, choice patterns are split up by valence (Win and Avoid Punishment) and Salience (reward vs. punishment, no reward vs. no punishment). Again, model M3 including only 1 parameter capturing the impact of Pavlovian biases appears to outperform the other 2 models M4 and M5 given the smaller rate of over- and underestimation of actual performance indicated by the smaller difference bars overall. The data underlying this figure and the plotting code can be found at https://osf.io/mcx36/.
(EPS)

**S6 Fig. Overview of parameter recovery.** (A) Scatterplots depicting the correlation (spearman coefficient) between the parameter estimates of the winning model M7 comprising 2 feedback sensitivity parameters for positive and negative outcomes, a learning rate as well as a Go and a Pavlovian bias computed for observed data as well as simulated data (parameter estimates averaged across $n = 100$ simulations). (B) Density distributions of parameter estimates of model M7 for observed data (green) and recovered data (blue). Dashed line indicates the mean of the distribution. (C) Scatterplot depicting the correlation (spearman) between the observed and averaged, simulated probability of making a go response, P(Go), and the probability of making a

correct response (left Go response for a Go cue requiring a left button press, a right Go response for cues with a right button press as optimal response, and NoGo response for NoGo cues). The data underlying this figure and the plotting code can be found at https://osf.io/mcx36/.
(EPS)

**S7 Fig. Mediation effect of positive feedback noise on task performance.** Mediation analysis indicated a significant partial mediation effect of feedback sensitivity for positive outcomes (rho +FB) on P(correct), i.e., the overall response accuracy when performing the task. Data and code to complete the mediation analysis are available at https://osf.io/mcx36/.
(EPS)

**S8 Fig. Cross-task association for noise parameters from the Go NoGo and Reversal learning task.** The scatterplots depict the association between feedback sensitivity for negative outcomes from the motivational Go NoGo task and the 2 noise parameters for positive and negative outcomes from the reversal task. Both correlation coefficients were nonsignificant (*p*-value > 0.1). The data underlying this figure and the plotting code can be found at https://osf.io/mcx36/.
(EPS)

## Author Contributions

**Conceptualization:** Annette Horstmann, Lorenz Deserno.

**Data curation:** Maria Waltmann, Nadine Herzog.

**Formal analysis:** Vanessa Scholz, Lorenz Deserno.

**Funding acquisition:** Annette Horstmann, Lorenz Deserno.

**Investigation:** Maria Waltmann, Nadine Herzog.

**Methodology:** Vanessa Scholz, Lorenz Deserno.

**Project administration:** Maria Waltmann, Nadine Herzog.

**Supervision:** Lorenz Deserno.

**Validation:** Lorenz Deserno.

**Visualization:** Vanessa Scholz.

**Writing – original draft:** Vanessa Scholz, Lorenz Deserno.

**Writing – review & editing:** Vanessa Scholz, Maria Waltmann, Nadine Herzog, Annette Horstmann, Lorenz Deserno.

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
