## [Editor Report · Decision Letter 0]

11 Jul 2024

Dear Dr Scholz, 

Thank you for submitting your manuscript entitled "Decision noise mediates the age-dependent development of specific reinforcement learning signatures" for consideration as a Research Article by PLOS Biology after our previous Open Reject decision.

Your manuscript has now been evaluated by the PLOS Biology editorial staff and an academic editor with relevant expertise and I am writing to let you know that we would like to move forward towards publication of your manuscript. The academic editor has assessed your response to the reviewers' concerns and agrees that all concerns have been addressed. 

We now need to do a couple of editorial checks for which we need you to complete your submission by providing the metadata that is required for full assessment. To this end, please login to Editorial Manager where you will find the paper in the 'Submissions Needing Revisions' folder on your homepage. Please click 'Revise Submission' from the Action Links and complete all additional questions in the submission questionnaire.

There are two items which I would like to suggest already now:

1) We would like to suggest a slightly modified title to increase accessibility: "Decrease in decision noise from adolescence into adulthood mediates an increase in more sophisticated choice behaviors and performance gain"

2) For the editorial blurb, we would suggest the following: "Learning and decision-making ability undergo substantial changes throughout development, with adolescents showing elevated levels of choosing suboptimal options ('decision noise'). This study shows that the development of specific and more sophisticated choice behavior in adulthood is linked to decreases in decision noise."

Once your full submission is complete, your paper will undergo a series of checks. To provide the metadata for your submission, please Login to Editorial Manager (https://www.editorialmanager.com/pbiology) within two working days, i.e. by Jul 13 2024 11:59PM.

Kind regards,

Christian

Christian Schnell, PhD

Senior Editor

PLOS Biology

cschnell@plos.org

---

## [Editor Report · Decision Letter 1]

29 Jul 2024

Dear Dr Scholz,

Thank you for your patience while we considered your revised manuscript "Decrease in decision noise from adolescence into adulthood mediates an increase in more sophisticated choice behaviors and performance gain" for publication as a Initial Research Submission at PLOS Biology. This revised version of your manuscript has been evaluated by the PLOS Biology editors.

As previously mentioned, we are likely to accept this manuscript for publication, provided you satisfactorily address the remaining data and other policy-related requests.

* Please clarify in the methods section whether the participants provided written or oral consent.

* DATA POLICY:

Regardless of the method selected, please ensure that you provide the individual numerical values that underlie the summary data displayed in the following figure panels as they are essential for readers to assess your analysis and to reproduce it: S2B and S5CDE.

* CODE POLICY

We expect to receive your revised manuscript within two weeks. 

*Published Peer Review History*

*Press*

Sincerely,

Christian

Christian Schnell, PhD

Senior Editor

cschnell@plos.org

PLOS Biology

---

## [Editor Report · Decision Letter 2]

2 Oct 2024

Dear Dr Scholz,

Thank you for the submission of your revised Research Article "Decrease in decision noise from adolescence into adulthood mediates an increase in more sophisticated choice behaviors and performance gain" for publication in PLOS Biology. On behalf of my colleagues and the Academic Editor, Matthew Rushworth, I am pleased to say that we can in principle accept your manuscript for publication, provided you address any remaining formatting and reporting issues. These will be detailed in an email you should receive within 2-3 business days from our colleagues in the journal operations team; no action is required from you until then. Please note that we will not be able to formally accept your manuscript and schedule it for publication until you have completed any requested changes.

PRESS

Sincerely, 

Christian

Christian Schnell, PhD

Senior Editor

PLOS Biology

cschnell@plos.org